# Management Attitude Shaping Cultural Sustainability in a Taxi Company—An Israeli Case Study

**Yaffa Moskovich**

Behavioral Science, Zefat Academic College, Safed 1320611, Israel; mosko777@gmail.com

**Abstract:** This research examined human interaction in a taxi driving company, analyzing the cultural influence of the management's social definition of its employees. In addition, this case study examined the relevance of "activity theory" through the perspective of the "iceberg model". Based on this model, the researcher explored the overt and practical layers of the organizational culture of the company as well as its hidden layers. The ethnographic research developed along qualitative lines: 28 interviews and 10 direct observations in the organizational setting. The research questions were: 'How did the owners' attitude shape the culture of their taxi company and how did their behavior influence cultural sustainability?' The findings portrayed a culture of conflict, driven by the owners' motivation to become rich quickly. The overt layers of the organizational culture included domineering managerial behavior, exploiting the drivers' inferior status, creating a high level of stress, and openly humiliating them. The drivers were deeply dissatisfied, which led to high rates of turnover. Moreover, analyzing the social interactions in the firm uncovered the hidden agenda of the owners (i.e., the covert layers), which was to amass a fortune in a short time at the expense of their workers. The analysis highlighted the damage this capitalist managerial attitude caused to the organizational sustainability of the company. The managerial behavior caused high driver turnover, which led to a constant shortage of drivers and instability in the company. By focusing on one taxi driving company in northern Israel, this research enriches the literature in the fields of social interaction, activity theory, organizational culture, and sustainability. This paper presents insights that stem from "activity theory", according to which managers can interact with their employees, overcoming mistrust and conflict, in order to enhance organizational sustainability.

**Keywords:** management; driver turnover; organizational culture; sustainability; taxi; cab

## 1. Introduction

This research focused on the ability of managerial attitude to shape organizational culture in one taxi driving company. Taxi companies have become a symbol of urban life. They illustrated change, mobilization, and modernization in the style of living [1]. This work enhances the interaction between research about sustainability and organizational culture by stressing the importance of the social factors among the three factors (financial, ecological, and social) that contribute to organizational sustainability [2–5]. The cultural analysis of this taxi diving company visualizes how the management exploited taxi drivers and how it affected the turnover of manpower in the company. This case study also provided an explanation of organizational behavior in taxi companies in general. [6–9].

The research questions were:

1. How management attitude shaped organizational culture of taxi company C?
2. What are the overt and covert layers in organizational culture in taxi company C?
3. How can we enhance organizational culture and sustainability in taxi company C?

Matzembacher and Meira [10] and Schein [11] have defined organizational culture as the normative glue that held an organization together. Schein's research also [11] adopted the iceberg model, which can discover hidden cultural dimensions behind the scenes. This

model analyzed the organization by beginning with the overt dimension, such as language, symbols, pattern of behavior, and artifacts, and tries to understand the hidden meaning of norms, values, and basic assumption. In addition, Schein's [11] model could indirectly provide insights about taxi accidents; although it was not the primary purpose of the current research. Previously, the literature about taxi companies discussed the process of drivers' unionization [12,13] and organizational factors that influenced the culture of safety and the rate of accidents. Most researches dealt with the question of how to reduce accidents rates [10,14,15], but ignored the influence of managerial attitude as an indirect cause of this dangerous phenomenon of company's organizational sustainability [3,16]. To fill this lacuna, the current study analyzed one taxi company from a cultural perspective, showing its relevance to organizational sustainability.

Borowiak [12], Mathew [13], Matzembacher, and Meira [10], Funke and Wolfson [15], and Bedi [17] researched taxi companies and focused on taxi drivers' work conditions, all of which contributed to high turnover among drivers: multiculturalism as a factor in internal conflicts between workers and managers [17], high risk, pressure, and low motivation [10,18]. This case study enriched the literature by supplying more details about taxi drivers' poor work conditions, which led to risky driving conditions and illness. This research analyzed the human relationship between drivers and their employer in the company. Previously, few research papers had taken this perspective. Most of the literature had focused on low wages and exploitive work conditions, which were a result of technological development and urban transportation changes [18–20]. This literature did not directly explain the source of poor work conditions, but instead, pointed to environmental, sociological, and technological causes in urban city lifestyle [1]. Moreover, the role of the owners as a source of these exploitive conditions has not received sufficient examination. In addition, the human resources factor has been an integral part of organizational culture, motivating employees and contributing to their commitment in the work environment [18].

The current case study focused on the conditions of managerial attitude, examining them in the taxi industry, filling a theoretical gap by pointing to the employer as a source of these conditions. This research dealt with social interaction among organizational members and its importance for the welfare of that organization. Moreover, this case study has contributed insights about the effect of troubled interactions between drivers and their managers: how the owners' attitude created a toxic climate and caused many drivers to quit [12,13,19,20]. Previously, few research papers had dealt with this internal perspective. The current research illuminated the difficult work conditions of taxi drivers from a managerial perspective and reported the mistrust between them and their employers in this company.

## 2. Review of the Literature

### 2.1. Social Interaction and Activity Theory

Social goals in an organizational arena can be achieved by interaction with other partners because each actor's action is interfered with by others "and since succeeding in one's aims is mediated by continual negotiation with other actors" [21] (p. 229). As Latour [22] noted, social interaction in an organization is an act of power and force, negotiation and authority, position and status that affects individual behavior. Social action of an individual cannot be understood without the cultural artifacts, evolution, and history that is embedded in a collective activity system. The uncertainty of dynamic life in social arenas affects the "cultural dope", which Latour [21] defines as the pattern of behavior, involving the cultural and moral life (i.e., norms and values) of each person's action, having an impact on the other actor in the interaction and within its environmental surroundings. The cultural analysis of human action defines its social structure, which puts boundaries on the social actor [23].

This research provides an example of how we can adequately apply activity theory and "understand the practical aspects of the instantiation of objects over time" [24] (p. 109).

Environmental work conditions are fundamental to understand human interaction in activity theory. This theory examines the behavior and motivation of social actors and how their cultural perspectives motivate their actions in a hierarchical division of work [21,22,25]. Social actors need to achieve organizational goals through collective activities. The individuals need to connect their own motives with their activities in work. Their activity is connected directly to their perception of the situation [6], which is a subjective interpretation of the real world. Their behavior affects other participants, who respond by creating a dynamic interaction.

According to activity theory, this process illuminates the development of flexible human interaction in working environments [23]. Practical aspects of the activity theory include various phenomena, such as: solidarity, strong organizational commitment, moral behavior, and mutual values of members of the organization. In part, these values rest on the four ground rules mentioned above [10,11]. This entails maintaining a positive work environment and a high level of respect in labor relations: concern for equal opportunities, safety, health, regulatory requirements, accountability, trust, and commitment [26]. This reinforces a positive atmosphere and strengthens consensus, while avoiding conflict. The supportive work environment lowers the workers' level of stress and increases honesty [20,27–29]. Employees have flexible schedules, permitting them to fulfill demands at work as well as deal with personal issues. This flexible environment enhances enthusiasm at work, organizational solidarity, and low absenteeism [25]. These positive social conditions augment workers' motivation and willingness to share information, which, in turn, increases organizational sustainability [2,3].

### 2.2. Cultural Sustainability and Managerial Attitude

Latour's [22,23] theory is connected to organizational culture, managerial policy, strategy, and style; each of which has the power to constitute the relationships in the organization and establish organizational routine, symbols, and norms. By doing so, they have tried to create rules, expectations, and values among managers and employees, in order to establish control and compliance of their workers to gain sustainability. The literature often calls these aspects the "triple bottom line" [2–5,26]. The frequent mention of these three dimensions in managerial reports and surveys about managers indicates their importance for formulating business strategy and specific policies [30,31].

There is a combination of four ground rules that can establish sustainability: "(1) a corporation working towards long-term economic performance; (2) a corporation working towards positive outcomes for the natural environment; (3) a corporation that supports people and social outcomes; and (4) a corporation with a holistic approach [that balances financial, ecological, and social well-being]" [32] (p. 40). These four principles are directly connected to the impact of organizational culture and its influence on labor relationships [24]. The interaction among the four principles can promote sustainability, shining a spotlight on cultural dimensions by visualizing the relationships among the members in the established environment. It has influenced the nature and behavior of the organizations' members and its study has provided a glimpse into the social order and culture-constructed reality of these organizations [1,21,22]. More specifically, organizational culture consists of a series of common principles and procedures that a given group has acquired as it addressed external adaptation to, and internal integration of, issues that the group found sufficiently valid for passing on as normative perceptions [26,30,31,33,34].

Recent research in organizational sustainability indicated that a strong connection between economic behavior, managerial attitude, and business strategy affects workers' production [20,26,27]. Sustainability and healthy organizational culture depend on quality management of human resources. Managers that are more sensitive to their employees' requirements and well-being display a paternal attitude towards them. As in other constructive culture businesses, the manager is like a mentor, encouraging and guiding his workers and teaching them to improve work output. A positive culture contributes to innovation, high productivity, and long-term sustainability [30,31,33]. Supportive management can

foster better interactions, harmonious familial relationships, better motivation to work, and better production [35]. Classical theorists, such as Marx, had already indicated the negative effect of managerial exploitation of workers. Trust and positive organizational beliefs enhanced employees' economic performance [36].

Alternatively, dishonest behavior of managers toward their workers and harsh work conditions produced organizational mistrust and low levels of production [24,25]. Managers can fire employees that reject managerial policy and replace them with new recruits more aligned with that policy. Authoritarian leadership demands compliance with the policies, but will also cause fear and mistrust among the workers. One of the outcomes of this troubled managerial attitude was the potential long-term weakening of organizational sustainability [5,16,26]. These problems exemplify Marx's critique of the labor relations in the capitalist system, but alternative organizational forms (such as cooperatives) offer better and more honest working conditions.

Management can achieve cultural sustainability by establishing more equal and just relationships with workers in a way that does not harm workers' productivity. Managerial habitus consists of a cognitive schema of knowledge, beliefs, and traditions; all of which influences the practical, personal behavior of all members of the organization. Obviously, managerial style influences the organizational culture; a supportive and democratic style will create less stressful labor relations. This was the case of several kibbutz industries, which avoided exaggerated salary disparities and offered mutually advantageous benefits to employees and managers. Organizational sustainability flourished under democratic leadership, as opposed to an authoritarian style [19,20].

Latour's [21–23] concept of social interaction is a complicated and complex process. It involves change and dynamism produced by the social actors inside and outside the organization. These actors can be a variety of groups, communities, and other entities. The sustainability of the taxi driving industry depends on external (e.g., environmental, international) and internal conditions, all of which influence social interaction between managers and drivers in the arena of work.

*2.3. International Features of the Taxi Industry*

Taxi companies have become a symbol of urban life. They illustrated change, mobilization, and modernization in the style of living. Over time, more taxi drivers have joined large corporate taxi firms. Big cities such as Mumbai, in India, needed quick transportation: they encouraged private investors, who shaped the workforce in the taxi industry [37]. This development had an impact on social, financial, political, and technological dimensions in modern urban life [21,38]. Along with trains, buses, and airplanes; taxis have played an important role in modern transportation. Taxi companies have reflected and facilitated the modernization of urban life, by providing faster social interaction in business, finance, and communication. This development has caused changes in the social structure among taxi drivers [1,21,22].

Taxi drivers have worked for small private businesses, but also for growing companies, which have accumulated great numbers of drivers. Some of the companies have supplied work to huge numbers of employees. In India, for example, kinship relationships have long dominated the structure of the taxi industry. Many owners of the companies have employed drivers belonging to their extended family. Nevertheless, self-employed taxi drivers who required independent licenses have existed alongside these family taxi companies [1].

An important aspect of the work conditions of taxi drivers has been the multicultural nature of many taxi firms, with many immigrants working as taxi drivers. This feature is common in developed democratic countries that are open to immigration: The United States, Germany, U.K., France, and Israel. The fact that taxi companies have increasingly heterogeneous work forces, something common to other organizations due to increasing migration in the world today, is important information from a sustainability perspective [2,3]. This was the case in Chicago, New York, and Philadelphia, where taxi drivers have been mainly immigrants, earning low weekly wages, ranging from USD four

to six an hour [12,13]. One American company consisted of German and Japanese members at all levels of the organization. This ethnic mix resulted in conflicts and difficulties in organizational communication between managers and ethnically diverse taxi drivers. The challenges that come with an ethnically mixed workforce has become global and it is relevant to other services, not just to taxi companies [17]. This multicultural nature tends not to exist in underdeveloped countries [16–18].

Another technological development in the taxi industry has been the emergence of online taxi services such as Uber, Lyft, Grab Car, and Go Car; using smartphone applications to connect riders directly with independent drivers (this technological development is more prevalent in America, Europe, and Israel than in Asia). This example of the growing adhocracy culture, or "gig economy", provides more independent and autonomous workers, contributes to a more flexible and convenient work environment, and promotes a sustainable organization. Nevertheless, there have been massive demonstrations and strikes by traditional cab drivers all over the world [18], protesting the "gig economy", which bypassed dispatchers and other administrators in traditional taxi companies and undercut the traditional rates, but another aspect of the "gig economy" is that the drivers' work environment has become increasingly temporary, volatile, and has reduced their income [18]. These exploitative work conditions caused American drivers to struggle for better conditions through union actions. They publicized their struggle and presented their collective demands during radio programs and in newspapers [19,20].

The literature about taxi drivers has addressed their strikes and other organized union actions, struggling for rights [13]. New York City taxi driver unions opposed the appearance of Uber, with its disruptive new technologies, bringing the drivers into conflict with the city government [12,13]. Uber was also a flash point in Philadelphia. The ATC (Alliance Taxi Cooperative) represented small taxi companies in that city and struggled against the exploitation of the drivers, who often existed below the poverty line [12,13]. This cooperative promoted the drivers' work conditions in the larger capitalistic system, demanding fair wages and encouraging a high level of class solidarity among the drivers. The ATC led the struggle against Uber, a company using new technologies that offered cheap transportation for the customer at the drivers' expense. These organized struggles were undertaken by taxi drivers in more democratic countries (i.e., in Europe and the United States), but were not common in Asian and African countries. Nevertheless, market forces have defined work conditions, without governmental interference, in most democratic and non-democratic countries [19,20].

### 2.4. Taxi Drivers' Risky Work Conditions

Worldwide, taxi drivers often suffer from risky work conditions and the lack of a culture of safety [39]. In Brazil, for example, drivers usually did not have a place for a rest break during the day, despite the high pressure of work, no place to eat properly, and long shifts of more than 11 h a day. This stressful environment increased the drivers' injuries, illness, and physical problems [40,41]. Thus, Brazilian drivers of taxis and motorcycle-drawn passenger vehicles suffer from health problems, are often involved in accidents, and lack basic social security in the workplace. Moreover, in Brazil, most taxi drivers are young males from a low socio-economic status. This phenomenon is common to other areas in South America, Africa, and Asia [39,42,43]. Risky work conditions also resulted from frequent conflicts between taxi drivers and clients over speedy arrival and the payment of fares. In addition, there were often conflicts among drivers competing for more clients. These factors intensified the risky and stressful work environments of taxi drivers [44]. One indicator of troubled work conditions has been the phenomenon of rapid turnover; drivers left the occupation and looked for another one. In Indonesia and other countries, this turnover stemmed from low individual motivation and low organizational commitment to retain taxi drivers [18].

High levels of stress, aggressive events, and negative attitudes towards taxi drivers caused them to look for other occupations. These problems make up another subject of

interest regarding sustainable workplaces/organizations. This subject and the problems of the gig economy have led to several outcomes in the taxi industry [15]. High risks at work, as an outcome of a minimal culture of safety, resulted in a great number of accidents in Iran. The causes of this low level of safety included economic pressure of the taxi companies on the drivers to supply quick service and a lack of trust in local police [39]. There were attempts to reduce stress, disease, and other factors exacerbating the high risk of injuries among taxi drivers in the United States [14]. A culture of safety was a major factor for the Canton Taxi Service, whose leadership style responded to the needs of the drivers and helped reduce accident rates [10].

## 3. Methods

The researcher became aware of and interested in Company C because she used this company for many years for traveling to the Zafet Academic College. This was a family business that began with two cabs and over the years had become a big firm. The success and development of the taxi company motivated the researcher to consider this firm for organizational culture research. During her taxi rides, she had heard many stories about the company from the drivers. She decided that a case study could provide insights about the organizational culture, management, and human relationships in this particular taxi company, as well as provide a point of reference for future research.

The current paper emerged from research during the five-year period from 2015 to 2020. It utilized the case study approach to examine the organizational operation of "Taxi Company C" [45]. The case study approach connected directly with the cultural interpretive method [10,11,46]. The research used ethnographic methods: a case study entailing 18 interviews and 10 direct observations. Ethnography is a method in which the researcher "penetrates another form of life" to "capture the richness of local cultural worlds" [44] (p. 209). The current work described and analyzed the ethnographic data of one case study by evaluating the internal views of the driver, by examining what people did and what they used in their social interactions. To complete the view, the researcher adopted theoretical and external perspectives [47]. The internal perspective would share empathy with the organizational member, but the external perspective would emphasize the analytical method replication and self-control of the researcher. This ethnographic research tried to understand the meaning of the culture by documenting everyday life. How did the drivers and owners perceive the reality of their taxi company? How did they interpret and justify their behavior? How did they define and construct their work and surroundings? What were the narratives they used for their everyday routine [48,49]?

Interviews and direct observations have been the core of ethnographic methods. These methods were adequate for this case study, which consisted of interviews [45,49]. The interviewees gave descriptions of the patterns of behavior in Company C; how drivers responded to the daily routine. The interviews supplied internal information about the experience of the drivers, the secretaries, and the dispatcher's assistant. To complete the data, the researcher also used direct observations, collected by the assistant researcher, who had worked in the past as a secretary in Company C and received permission from the owners to take notes and summarize her observations.

The external and internal perspectives have been supplied by the assistant researcher, who had previously worked in the taxi company, but at the time of the study was working in the Zafet municipality. Her previous connection with the taxi company helped her to conduct the interviews. Because she had left this company two years previously, she could assess the firm with an objective perspective.

It is important to note that the constructive qualitative researcher is an integral part of the study, as an observer or interviewer. Patton [50] (p. 121) declared: "To understand the world, you need to become an integral part of the research and at the same time to be separate, to be in and out". The constructive researcher needs to identify with the interviewees and empathize with them [51]. One challenge of qualitative research is finding the golden path regarding interviewees: between involvement and empathy, on

the one hand, and criticism and social distance, on the other. It requires the researcher to be reflective. In this case, the assistant researcher achieved reflectivity through interaction with the lead researcher, who gave guidance in conducting the interviews, keeping enough psychological distance in order to avoid research bias. To enable the processing of data, the researcher built analytical categories, using the iceberg model for guidance. Cognitive analysis and mutual thinking helped overcome subjectivity and empathy [52].

These data enriched the case study and provided detailed descriptions of the drivers' schedules, the stress in the work place, as well as the conflicts and disputes among the participants in the organizational culture of Company C [6–9].

### 3.1. Data Collection

Interviews gathered data from three groups: drivers, owners (management), and administrative workers. Each group had different features and the information came from several points of views. For instance, the drivers were a heterogeneous group, often in a state of flux; some worked full-time; some worked part-time; some were veteran employees; some were short-term workers while looking for better jobs or the opportunity to become independent taxi drivers. Nevertheless, all of the drivers supplied information about the destructive organizational culture. The management was stable over time, the founders of the company described their goals and aspirations, which were important in analyzing the culture of Company C. The last group consisted of administration workers, who provided information from the office, where they gained information that the drivers were not able to gain. This knowledge completed the first and the second research groups. Each group supplemented the other two in forming the overall picture of the organizational ethnography of the company. All of the informants contributed to delineating the linkage between organizational culture and sustainability in Taxi Company C.

As noted, the taxi drivers made up a very diversified group. Some of them were pensioners who wanted to earn some extra money during their retirement, or were bored staying at home. This group included educated drivers, some of whom had been Israeli army officers, or had positions in the Israeli postal service, in the police, or in other bureaucracies. Other drivers were younger. This second group usually consisted of uneducated workers who had no opportunities to find a better job. Some of the drives were "between jobs" and worked temporarily as drivers until they found a better position. The drivers were also ethnically diversified, consisting of secular Jews, ultra-Orthodox Jews, Christian and Moslem Arabs, and Druze. Some of the Jewish drivers were new immigrants to the country. Most of the drivers in the company were male; although one of the interviewees was female. The owners were a husband and wife team. Most of the interviewees reported that the female owner was dominant in running the company: she had knowledge in finance, while her husband was only a driver in his former occupation. The last group was the administrative workers that included a secretary, a dispatcher, a sub- dispatcher, and two bookkeepers. Another way to divide the group of workers was by their seniority and age. Some, in the case of the dispatcher and sub-dispatcher, had worked a long period at the firm, while others were newer and younger employees.

Drivers, employers, dispatchers, and other employees participated in eighteen ethnographic interviews. The ethnographic interview was an open interview, more like a conversation, to establish a relationship of trust and participation [53]. In this kind of interview, the informant usually discussed the organizational culture. This type of interview was relevant to this study because the purpose was to explore the hidden layers of that culture [49,53,54].

The researcher organized the interviews into two rounds. The first round was during the period of 2019–2020 and included 18 interviews (five conducted by the assistant). To complete the data, the researcher added 10 interviews (three conducted by the assistant) during the second round, in the summer of 2022. During that second time period, the researcher interviewed another sub-dispatcher and nine additional drivers, three of whom had retired from the firm. The data collected in the second round were similar to those in the first round. Most interviewees complained about the difficult conditions at

work and about the owners humiliating them. Only one driver was satisfied with the work conditions.

The ethnographic interviews took place in several arenas: in the company's office, but also during taxi trips [50,54,55]. The researcher was a lecturer in an academic college and used the taxi service to travel to work and return home (the college routinely worked with this taxi company, which drove the lecturers to the college on a regular basis). The researcher had numerous conversations with the taxi drivers and also heard drivers communicate with their employers during the trip. The researcher interviewed 10 drivers, the two owners of the company, two dispatchers, an assistant to the dispatchers, two bookkeepers, and a secretary.

Table 1 presents the details of the workers interviewed at Taxi Company C.

**Table 1.** The Workers Interviewed at Taxi Company C.

| Position | Seniority at Work (Years) | Age and Gender | Ethnic Group |
|---|---|---|---|
| Owner | 1 | 50 female | Jewish |
| Owner | 17 | 55 male | Jewish |
| Secretary | 3 | 30 female | Jewish |
| Dispatcher | 17 | 60 male | Jewish |
| Sub-dispatcher | 10 | 40 male | Jewish |
| Sub-dispatcher | 1 | 45 male | Druze |
| Bookkeeper | 2 | 35 female | Jewish |
| New bookkeeper | A few months | 38 female | Jewish |
| Minibus driver | 1 | 50 male | Arab |
| Temporary taxi drivers | A few months | 25−30 male | Arabs |
| Veteran driver | 5 | 60 male | Jewish |
| Taxi driver | 2 | 40 male | Ultra-Orthodox Jewish |
| Taxi driver | 1 | 46 male | New immigrant |
| Taxi driver | 1 | 50 male | New immigrant |
| Taxi driver | 2 | 42 female | Jewish |
| Assistant (helped the sub-dispatcher) | 2 | 38 male | Druze |
| Taxi driver veteran | 15 | 70 | Jewish |
| Taxi driver veteran | 2 | 68 | Jewish |
| Taxi driver veteran | 1 | 70 | Druze |
| Taxi driver | 5 | 50 | Jewish |
| Taxi driver | 1 | 48 | Ultra-Orthodox Jewish |
| Taxi driver | 3 | 55 | Moslem Arab |
| Taxi driver | 2 | 40 | Moslem Arab |
| Minibus driver | 2 | 36 | Druze |
| Taxi driver | A few months | 38 | Ultra-Orthodox Jewish |

Most interviewees requested anonymity; given their status in the taxi company, they did not want to reveal their identity. It was noteworthy that many of the drivers were afraid to talk about certain matters. They thought that providing information about Company C could cause them to lose their work [56]. The researcher acceded to these requests, which was useful because in ethnographic work interviewees are more willing to talk openly about their lives, and because this work was ethnographic, the researcher brought citations



from the field to visualize authentic organizational culture. The researcher recorded some of the data immediately, writing as the interviewee spoke; but in other cases, the researcher summarized and transcribed the statements after the interview.

The specific topics of the interviews changed according the interviewee's position in the company: drivers, office workers, and owners. Nevertheless, all the interviews focused on the human relationships in the company. Each interview lasted an hour. Because the drivers had a lot of complains about the owners it was not difficult to fill the hour. In general, it was easy to talk with the drivers and the conversation was open and free. They discussed their work conditions, the reasons for the turnover phenomenon, and how they felt in the company. The questions posed to managers and assistants focused more on the managerial style in the company, their views, their aspirations, and their expectations regarding the drivers. The topics for the owners were different and concerned their managerial beliefs, their motivation, their current aspirations, and their expectations about the future. Other office workers described their work experience, feelings of satisfaction or dissatisfaction, and their duties in the office.

Direct observation was a central method in the study's qualitative approach, with the researcher noting the factors that constructed the researched environment. This was the initial process that helped map the researched arena, deciding what the central focus was and how to progress in the study [47,48]. The researcher collected "rich data" that contained descriptions, conversations, and artifacts. Each item helped complete the analytical picture [56].

Ten direct observations took place in the last year of the study, in 2019−2020. A research assistant conducted the observations, each of which lasted three hours; a total of 30 h of observations. This research assistant watched the workers and summarized the conversations that took place. The research assistant also described the atmosphere in the company. For instance, the drivers and other employees seemed afraid of the female owner. Although a husband-and-wife team owned and managed the company, the real boss was the wife. The observations described the activities and physical artifacts in the firm: the work routine, coffee breaks, pictures on the walls, as well as the language and linguistic symbols that were common in the company. The researcher collected all the data and analyzed them, using interpretive methods, to understand the meaning of the language, as well as the jargon the drivers and employers used among themselves. The data from the observations helped to identify the obvious layers: language and patterns of behavior that were part of the daily routine. The researcher then analyzed the deeper, hidden layers of the culture: norms, values, and basic assumptions.

Company C was the subject of this study because it had existed for a long period of time, and thus could supply more data about the work conditions of taxi drivers than examining younger taxi companies. Its existence over time facilitated cultural analysis, allowing the discovery of the overt and covert layers of the company's organizational culture. In addition, Company C was a big company with a large number of drivers, which could more clearly present one example of work conditions in Israeli taxi companies [21].

Validation and reliability in the Company C case study [45,46]: The collection of qualitative data proceeded step-by-step over a one-year period. In addition, the assistant researcher had previously worked in the firm for several years and her observations clarified the findings. The researcher and her assistant spent considerable time in the research field, internally validating and ascertaining the reliability of the data [49,50,53,57]. They compared each piece of information, from each interview, looking for similarities and differences among them [53]. The researcher validated the findings by triangulation; at least one additional source of information confirmed each bit of data. For example, drivers validated the statements of other drivers. In many cases, other informants from the office (secretaries and assistants) validated individual driver's statements. Moreover, direct observation by the assistant researcher validated some of these statements. In addition, various office workers validated the owners' statements. In the same way, veteran administrative workers, such as the dispatcher and sub-dispatcher, described the events

and the managerial attitude in a similar manner. Thus, the similarities among the pieces of information were very strong, with a few minor differences; so, the data from the interviews supplied a very clear picture of the organizational culture. Moreover, the reliability was supported by: "including rich and thick verbatim descriptions of participants' accounts to support findings" [57] (p. 1). Thus, the article has many verbatim descriptions from each of the three groups of participants.

　　Recovery reliability: In case studies it has been difficult to achieve recovery reliability [45–47] because each data point emerged at a different time and place. The circumstances of the data collection frequently changed. The drivers participated in interviews during their journeys or outside the office. At different times, the direct observations supplied similar descriptions of routine behavior of everyday life in the office. Nevertheless, some episodes were different and focused on special issues, such as an urgent call for a taxi, an incident when the driver was late, and when a driver did not appear for his shift, etc.

　　The second round of interviews contributed to the reliability of the data and analysis. After two years, the conditions were a little different than in the first round. The firm had grown: operating over sixty cabs. In addition, the COVID-19 pandemic had subsided. Nevertheless, the researcher heard similar accounts as in the first round [50–52].

*3.2. Data Analysis*

　　The ethnographic perspective critically scrutinizes previously unexamined beliefs and assumptions [51]. The current study explored the management's business perspective and found that most company's workers accepted these views. Given the importance of researcher reflexivity in ethnographic research, the researcher made a special attempt to understand the importance of the different perspectives expressed by the participants. To this end, the researcher engaged in all routine activities the members of the firm proposed. Members of the company organized daily activities, meetings, and interactions with other drivers, employers, and dispatchers to facilitate understanding of the structure of realities at the taxi company [6,52]. The goal of the data analysis was to understand the cultural reality in order to "make sense of what is going on in the scenes documented by the data" [57] (p. 209).

　　Once the researcher completed the interviews and analyzed the direct conversations, the researcher proceeded to code information into relevant categories arising during analysis. Categorization appeared to be the most likely means of connecting the cultural behavior to the theoretical foundations of social construction [6,7]. The researcher assigned each data item to meaningful categories, facilitating the interpretation of all information and the construction of a narrative about the firm's organizational life. The iceberg model [10,11] was the data analysis method for the coding (categorization) of the data points: the researcher first analyzed the overt dimensions of the culture (rituals, symbols, language, and patterns of behavior) and then went on to uncover the hidden layers of the culture (norms, values, and basic perceptions) [32–36]. Grounded theory guided the categorization as the researcher connected narratives, stories, and organizational events of the specific taxi company's reality to theories of organizational culture and sustainability literature regarding taxi companies [2,3,10,11,45,47,58]. Research method was visualized in Figure 1.

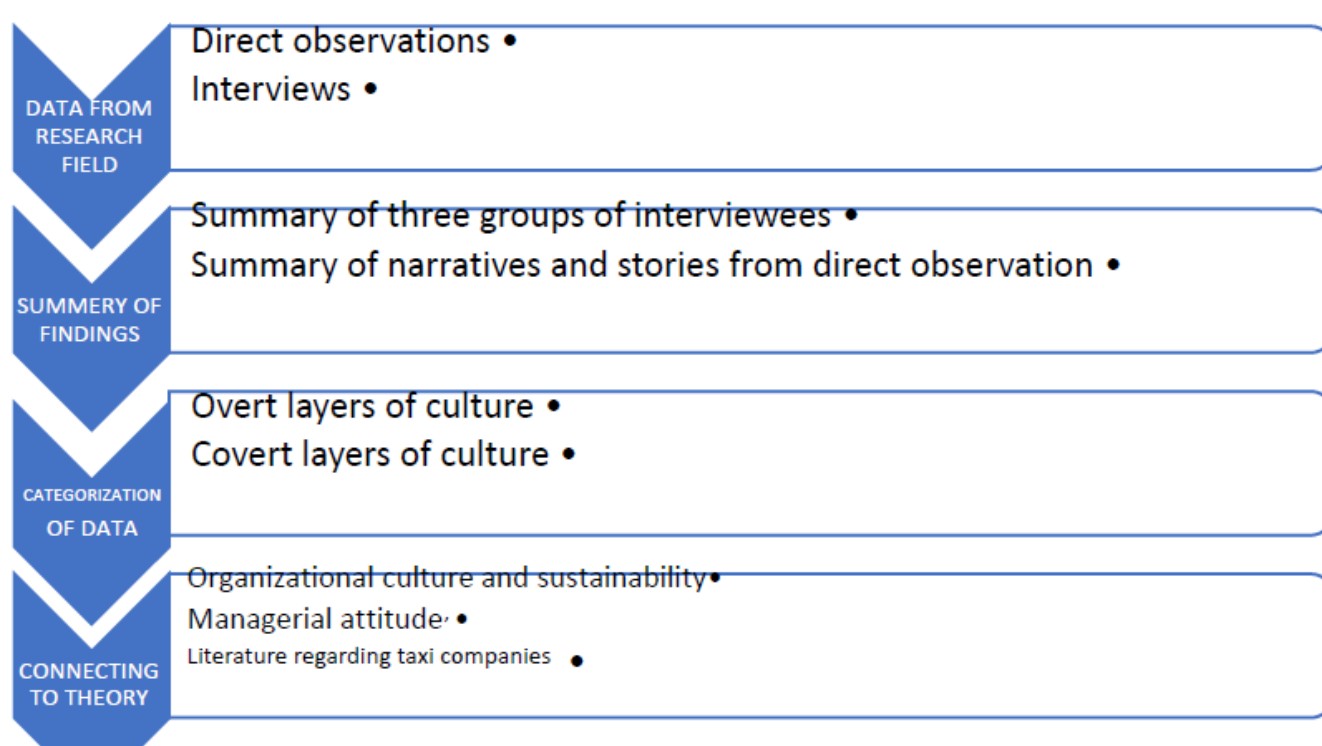

**Figure 1.** Research graph.

## 4. Findings

The iceberg model [10,11] provided the framework for presenting the findings of the current research. The analysis began with the overt dimensions of organizational culture (see below in Section 4.1) Nevertheless, most of the analysis dealt with the covert dimensions, expressed in categories such as: various types of norms, conditions of work, turnover of the drivers, and the company's treatment of the drivers. These were the outcomes of the cultural norms. The company's treatment of the drivers and its managerial style illuminated the hidden values and basic assumptions of the company. These findings visualized the problematic human relationships and the exploitative conditions. Critical analysis of the firm's organizational culture discovered the real nature of this company [16,25].

### 4.1. Overt Cultural Layers of Organizational Culture in Company C

This section presents the goals and aims of the taxi company, as reflected by the overt items that its members and customers were aware of: historical background, organizational artifacts, patterns of behavior, and division of work.

### 4.2. Historical Background

In 2003, seventeen years prior to the study, a husband-and-wife team established and began operating "Company C", a small car wash with two cabs. This taxi firm operated in the city of Zefat, in northern Israel. The family rented a small office with the help of two friends, who later became the company's principle dispatchers. With the expansion of the business, the firm opened another line within the company named after the owners' daughter. Originally, the wife's father and other relatives worked as taxi drivers, but as the company grew larger, most of its taxi drivers were not members of the founding family. One of the dispatchers reported that, in contrast to its meager beginnings, the company had more than 50 cabs, 10 regular minibuses, 10 minibuses for handicapped people, and 14 minibuses of other types. During the research time, the company hired 86 workers, including: the family owners, the main dispatcher and his assistant, a secretary,

two bookkeepers, and an officer in charge of the vehicles' security and the drivers' health. The company worked with 30 subcontractors, who owned their private taxis, and 50 taxi drivers, who worked as direct employees of the firm.

The company provided transportation services to school children, older students in the northern area of Israel, factory workers, and workers at the local hospital. The firm had regular arrangements with institutions in the northern area in Israel: most of the minibuses worked half a day, driving children to schools or workers to hospital and factories. The taxis of Company C delivered postal packages all over the country. They also delivered blood tests to laboratories from prisons and hospital in the northern area.

### 4.3. Patterns of Behavior in Taxi Company C

At the time of the research, the dispatcher and sub-dispatcher organized the work by dividing it among the drivers. The assistant helped the sub-inspector by replacing missing drivers as well as receiving and processing calls for taxi services. The vehicle officer was responsible for the maintenance of the cabs and renewing their taxi licenses.

The duties of the secretary included typing letters to customers, typing all sorts of reports, and delivering calls to taxi drivers. The secretarial work also included collecting the bills and vouchers for gasoline, as well as sending bills to customers at the end of the month and collecting payments. In addition, the secretary was also responsible for the maintenance and cleaning of the building. The secretary said that she did everything that the office needed. In fact, she noted that the female boss had called her a "kolboinik", referring to a deep bowl set on a dining table (mostly in communal dining halls in old-time kibbutzim) to collect the left-over scraps. The secretary and the bookkeepers sat together in the office and helped each other. One of the secretaries frequently complained:

I am a trained secretary. Instead of working in the field of my expertise, the female boss constantly gives me unprofessional and strenuous petty tasks. I feel that my knowledge is not used properly. If I have an opportunity, I will probably look for another position in another place.

The company worked with several artifacts: computer software, storing all the information and presenting it on two screens. The first screen showed all the trips, the schedule, who ordered the trip, the name of the driver, the prices, and other pertinent information. The second screen presented a GPS software tool called "Pointer". The owners had inserted a digital chip for Pointer in every car so they could monitor the car's activities. The software showed the pathway of every driver, listed the passengers, and if the car was moving or stationary. In the office, there was a map that presented this information. This system helped the owners to know what was happening in the field. If a customer complained about a delay, the owners could see where the drivers were.

### 4.4. Division of Work among the Drivers

The dispatchers divided work among steady workers and temporary drivers. The steady workers were satisfied with the additional income from driving. One of the drivers said that steady drivers had regular routes, but many of the drivers worked according to temporary arrangements and needed to take every trip that the owners requested of them. These temporary drivers had irregular arrangements: delivering blood tests from the hospital, delivering random packages, or random trips. As one of the drivers said: "I drive when and where my boss asks me". Young drivers drove to several different places; the dispatcher gave them trips to the center of Israel.

On Saturdays, Arab drivers took over all transportation in the company, while the Jewish drivers rested during Sabbath. The Arab drivers usually took elderly people from Arab villages in the area to visit their relatives in senior citizens' homes.

## 5. Cultural Layers in Company C

One of the goals of this research was to discover and analyze the various layers of the organizational culture of Company C; some were obvious; some were hidden. Examining

the various norms and values of the company allowed the discovery and interpretation of its work conditions and managerial style.

*5.1. Managerial Policy: Shaping Work Norms for Drivers in Company C*

The Israeli Ministry of Transportation mandated every taxi company to require drivers to have government-issued licenses for public transportation. Companies could only hire new drivers after they had passed a driving course and had received this license for public transportation. All the drivers in Company C had passed the course and earned that license.

Various norms were structured by the company' owners:

1. Drivers had to come to work alert after resting and sleeping. One of the drivers stated that the dispatcher had told him: "Go rest, go to sleep. You need to work early in the morning". During their work, the drivers had to take breaks to refresh themselves before continuing.

2. Drivers had to inform the company if they were not available to work for personal reasons or holidays. During one of the observations, a driver informed the dispatcher that he needed vacation days to take a trip with his family during Christmas time.

3. Drivers had to clean their vehicles and needed to tell the boss if there was a problem with the car. During one of the observations, the dispatcher yelled at an ultra-Orthodox driver who entered the office: "Clean your cab a least once a week! This is not a garbage dump!"

4. The owners demanded that the drivers talk quietly while in the office. The female boss did not allow them to talk about their driving. The drivers came to the building only to deliver envelopes of money.

5. Drivers could not stay in the building except for making coffee or meeting the owners in a business meeting. During one observation, the dispatcher was chatting with one of the drivers who had come in, and the female owner shouted at them: "Quiet! I don't want to hear talking". She then addressed the driver: "Do you need something?" He answered: "I came to deliver the money". She replied: "Give in the money and return to your work. You don't have any business to be here". The driver took his coffee and left the office.

6. The owners expected the drivers to work long hours in order to keep the car occupied. The average shift was about 12 h each day. This regulation was quite difficult for the drivers; especially the older ones, who expressed their discontent and told the researcher that they were looking for a less stressful place to work.

*5.2. Norms of Driver Payment in Company C*

The drivers had to deliver the fares they had collected every day as well as deliver bills for reimbursement for their expenses. This delivery was a ritual. A driver would come to the office to hand in his envelope with his fares and asked the owner for more work. Despite the clear regulations, the drivers occasionally disputed the payment with the dispatcher, complaining that they had not received the money for all of their trips. When they delivered the money, the drivers wanted reassurance the office had listed the payment.

The owners complained when a taxi was not operating because they lost money. One dispatcher explained: "The taxi is like an industry; it produces money. You need to operate the taxi 24 h, seven days a week. Only in this way can you earn money".

A very small group worked with monthly salaries. This group consists of veteran workers who had been members in the company for more than ten years. The company retained this group of drivers because most of them received pensions from their previous work. Most of the drivers received 30 percent of the fare from a trip. The drivers had to pay for gasoline, but the owners covered maintenance expenses for the vehicles. The drivers had a lot of complaints about this system of payment and felt that the owners were exploiting them. One of the drivers said: "The owner of the company works on a low payment rate. To win a tender from a potential organizational client, the firm needed to

compete with other transportation companies, so he (the owner) works on really very little money and I work on nothing". Company C competed for tenders in the local municipality, Zefat Academic College, the regional hospital, and various factories. Usually, the owners won contracts by offering lower rates than other companies. The low rates gave the owners a lot of work, but the drivers felt that the owners' profits were at their expense. The researcher heard this complaint often and some drivers stopped working for the company. These drivers either transferred to other companies or became independent drivers. Some of the drivers rented their taxi to the owners, who supplied them work. One older driver was very angry at the owners and often complained that the owners consistently delayed paying his wages. At the time, this driver depended on income from the company, but frequently said: "They are criminals and cheaters. I am looking forward to the time that my condition will improve and I won't work with them". By the end of the research period, this driver had stopped working for the company.

*5.3. Managerial Norms concerning Work Conditions and Treatment of the Workers*

Most of the drivers and office workers were discontent with the conditions of their work at the taxi company. A few, though, were satisfied. These satisfied employees enjoyed the freedom that the work offered them. Work hours were flexible, they traveled, and met people. A veteran driver, who had worked in the company for more than ten years, also said that he liked the work. He liked the owners and the nature of the work, which gave him the opportunity to diversify the places that he was traveling to and he also liked his regular clients in the taxi. Another veteran driver said that the owners had given him easy work with short shifts after he became ill and suffered a heart attack.

The secretary also mentioned the flexible work hours, which allowed her to combine work with domestic obligations, such as taking care of her children. The ultra-Orthodox drivers had extra money by driving. Although they received money from their yeshiva (religious educational institution for adult men), the work provided them more income and flexible work hours. They could study Torah (the central Jewish holy document) and when they were available, they worked some hours in the evening or at night. However, most of the drivers did not like to work for the company; they worked there because they had few alternatives. The work in Company C entailed a great deal of stress. One of the drivers complained about the pressure to work long hours because the company had a shortage of drivers. Another driver talked about the pressure he felt. When he had been ill, the owners forced him to work because of the shortage of drivers. He noted that once he was so tired that he got into an accident. Another driver complained that the owners had forced him to work long hours, despite his recently suffering from heart illness. He was so angry at the owners that he decided to resign. Obviously, work conditions were naturally demanding and complicated, but the treatment of employees by the owners made the situation even more difficult. Drivers interviewed in the second round said they earned very little money. One of them said: "It is not enough for anything; everything is very expensive. I am looking for another driving company that will give me a better salary".

Drivers, dispatchers, and other employees were unsatisfied with the owners' treatment, with more complaints aimed at the attitude of the female owner. The entire atmosphere would change when she was present, with a lot of tension in the air. The following incidents demonstrated the working relationships that the researcher observed. The female owner once shouted at the dispatcher in the office: "Why do you send this driver? He is a beast; you can't rely on him". That owner insulted a driver: "Come here, dirt! Let's see how you will behave. What you are going to do?" This language expressed the treatment of the female owner towards the drivers when they were late or did not accomplish their missions. Drivers complained about the low wages they earned. The owners did not like to discuss salaries with the drivers. If someone began to talk about the issue, the female owner would change the subject and told the driver "go rest, go to sleep" (direct observation). Similar humiliating episodes appeared in the second round of interviews.

The drivers did not want to talk with the owners about their wages because they were aware of the female owner's unpleasant reaction. Her attitude and rude language did not encourage the drivers to interact with the owners. Drivers usually tolerated this verbal abuse until they found alternative employment. The constant control, criticism, tracking, and surveillance gave the workers the feeling that the owners mistrusted them. The drivers did not like the monitoring and interference in their trips. On more than one occasion during the researcher's trips to work at the Zefat Academic College, the researcher heard drivers arguing with an owner (direct observation). Once the owner asked: "Why are you taking this route?" The driver answered: "This is a better and a shorter way". The owner replied: "Change your route!" In the end, the driver convinced the owner to accept his chosen route. After he ended his conversation with the owner, he complained and said: "It is very difficult to work like this. The owner constantly intervenes in our trips". Because the monitoring system provided detailed and real-time information about the drivers, the owners used it effectively to control the drivers.

Even the dispatcher, a veteran in the company, complained about the negative environment caused by the owners. The dispatcher noted that "he handled everything" and routinely saved the owners money and time. He solved problems with the drivers all the time, yet the owners frequently complained that they could not rely on him, or on anyone else in the company. This was an insult to his loyalty and professionalism at work. On the other hand, the female owner put the blame on the workers who, according to her, needed to show more involvement and responsibility. During an interview, she complained about the constant need to supervise her workers because she could not rely on them. She claimed that they played the 'small head' (Israeli slang for someone who did not take the initiative in a situation), a situation which she found extremely tiresome.

The resultant climate in the office was unpleasant. One of the minibus drivers said: "She (the female boss) constantly feels that we are stealing from her. She doesn't pay as she should" (direct observation). On the other hand, when the male owner was pleased, he showed it in his language: "How are you?" "Good soul, I love you". He referred to God in his positive phrases. His language was laudatory, with a lot of slang, such as "darling", "really love you", and "pure soul". He was pleasant when a driver delivered the day's envelope with fares to him, hugging the driver and saying: "I really love you, you are great" (direct observation).

The dispatcher, who had worked in the company for many years, expressed another view. He saw the owners as family, saying that the owner loaned money to the drivers. Moreover, when the police once arrested a driver, the male owner came with a lawyer and freed him from jail.

### 5.4. The Outcome of Managerial Attitude: Worker Turnover

Because of the hard work conditions and poor employer–employee relations, the company experienced a high rate of turnover. Most of the drivers did not like the owners, so when they had problems, they preferred to talk with the dispatchers. This would often lead to dysfunctional situations and the desire to leave the company. The office workers did not tend to stay very long in the company either.

One dispatcher noted that it was difficult to find and retain cab drivers because there was a lot of competition from other companies. The drivers said that the work was difficult and very stressful because the owners are never satisfied. As a result, the drivers constantly threaten to move to other companies. The bookkeeper had decided to leave the company and the dispatcher accepted her resignation the previous day. The secretary mentioned that the previous year, she (the secretary) had an argument with the owner, but she stayed and continued to work, but other workers found work in other companies.

### 5.5. Managerial Values and Basic Assumptions Reflected in the Managerial Style of Company C

The owners managed the company in an autocratic way; the female owner checked everything: daily work plans, bills, and outgoing emails. She constantly consulted with her

husband on these matters. She said during an interview that she is very tired of controlling all of the work including trips, vouchers, and receipts. In addition, she noted: "A cab that is not working causes us to lose money. I don't accept that a driver refuses to make a trip because it is not worthwhile".

A female driver pointed out that workers in the firm were afraid of the female owner, saying: "Everyone in the office says that she runs the business. She knows everything; she is involved. She is also aware of things because of the cameras that are everywhere. She is very strict and does surveillance of all the workers". The secretary added: "I saw her collapse because she worked very hard, even she cried from despair and prostration".

The decision-making process in the company does not consult drivers or take their needs into consideration. Nevertheless, the secretary noted that the expectations placed on the drivers were extensive. Sometimes the drivers had to make quick changes. The secretary said: "They expect the drivers to obey, not to initiate or be creative". The assistant said: "They want us to blindly obey. They don't take our opinions into account".

The owners believed that the success of the company was an outcome of their constant involvement. The female owner said: "Investment, continuity, and credibility–this is our secret of success. That was the way we established the company. This way it will continue: only by strict and professional work." The male owner was a hard worker. He stayed in the office for long hours during the day and, if needed, also at night. One of the drivers complained that the owners were very greedy.

The vision of the owners was to enlarge the company and to become rich in a short time; this was the basic assumption of Company C. Over the years, the company has become one of the biggest taxi firms in northern Israel. The male owner said: "I want to provide the best service, the fastest–efficient and pleasant". The dispatcher said: "The owner dreams of coming to the office and not having to work like a donkey". Nevertheless, the dispatcher observed: "The owner already fulfills his dream, but they are losing their lives in the business. They don't have any pleasure, relaxation, or fun from his prosperity" (direct observation).

## 6. Discussion

This case study, about the taxi drivers' workplace, was an example of using "activity theory" to examine the practical aspects of human interaction [24,25] and the cultural motivations of the social actors [11,26,30]. The pressure and stress that characterized the taxi drivers' working conditions, with the capitalist environment in general, and the ambitious owners specifically, illuminate the cultural aspects of theory in action [11,23]. Each actor had a different motivation in the social arena [21]; the main goal of the owners was to enhance profits, while the drivers hoped for reasonable working conditions. Capitalism caused all of the social actors to become ambitious and desire wealth in the short run. Society's "definition of situation" [6] forced social actors to adapt their behavior to accomplish this goal.

This case study provides an example of the common theme of businesses creating exploitive working conditions. Capitalist ethos and morality cause problematic behavior [13] that encourages social actors to accumulate wealth without considering ethical norms [6,21–25]. On the other hand, capitalist environments provide other aspects, such as: achievement orientation, professionalism, liberalism, and fair working conditions. Although capitalism motivates social actors to prioritize profit, competition with similar actors (i.e., alternative employers) will theoretically improve the working conditions. In these alternative firms, managers maintain a stable work environment and a rewarding relationship with their employees. Capitalism is complicated and heterogeneous; managers can calibrate their behavior by "definition of situation" as an outcome of history, societal development, and cultural beliefs [6,12].

This research illuminated the problems of the capitalist system in which the taxi industry has existed. Taxi drivers have struggled against technological developments that undermined their traditional work and against worldwide low salaries and poor conditions.

The current research has augmented the information about the taxi drivers' poor work environment by pointing out the central role of the owners, who often treat their employees like slaves. The previous literature presented a meagre discussion about the perspective of the employers in the taxi industry [1,12]. The historical development of cities has provided numerous opportunities for investors to enter the transportation sector and make easy money [21].

The problematic social interactions appear in the overt and covert layers. As Latour claimed [21,22], social interaction involved force, authority, and power relationships. In this case study, the hierarchy in the organization and its division of work led to conflicts between the owner and the drivers [21,22]. The unbridled, superior power of the owners created the employees' difficult working conditions and led to the phenomenon of frequent worker turnover [12,13]. The researcher directly observed Latour's concepts in the taxi company, corresponding to Schein's [11] iceberg model, which facilitated the analysis of the organizational culture of the company. The owners created rules and regulations to control the workers and to make the maximum profit from their work [32,51]. Their authority and power interactions appear in each episode in the work arena [21–23]. For example, the female owner did not allow drivers to rest in the office. Her desire to hide the information about revenues was probably the motivation for demanding drivers to be out of the office. Activity theory [24,25] explains the practical norms and patterns of behavior in the firm, which expose the superiority of the owners and their ability to construct daily routines to their advantage.

In the overt dimension, the owners asked the drivers to deliver the fares at the end of their shift. To ensure full delivery, they surveilled the drivers using electronic devices. In addition, maintaining the norm of long shifts caused stress and burnout. This heavy-handed behavior reflects the covert dimension of exploiting the drivers as much as possible in order to maximize profit. The owners' basic goal was to become rich as quickly as possible. Their basic assumptions about the company flowed from this goal. This underlying and covert greed manifested itself in the overt dimension as the owners' ambition to provide the public with a quick and professional taxi service.

Here, Latour's [21,22] theory highlights social interaction connected to cultural sustainability. In addition, management attitude theories offer a comprehensible analysis according to the iceberg model. According to these models, relationships in Company C stemmed from the conflictual institutional logic of capitalism, that reflected the opposing needs of the owners and their employees. In this case, the owners succeeded in exploiting their workers, while the workers unsuccessfully tried to resist. The result was an autocratic managerial style [52], producing mistrust and animosity between the owners and their workers [42,59–61]. In Company C, only the owners constructed reality on their terms. They reinforced daily that they considered workers untrustworthy and unreliable. The owners accused the drivers of being cheaters, who wanted to rob the owners' wealth. This negative attitude produced a cycle of negative interactions, which constructed a reality of suspicion, animosity, and mistrust [6,7]. This negative environment, consisting of conflictual and harmful social interactions, is also visualizing "activity theory"; how a negative managerial human relationship became a threat to organizational sustainability [2,3,5,24,25].

This was an example of human social interaction [6,21–23]: practical managerial behavior and its effect on the drivers expressed the dynamism of social definition. There was a cycle of responses. Humiliating managerial behavior and exploitive salaries caused the drivers to resign and look for work in other taxi companies. As a result, the owners recruited new drivers, who, in turn, left the company after a short and negative experience. This harmful cycle was continuous because the owners behaved consistently in ways described in the research data [24,25] (pp. 17–22).

Despite the misleading overt dimensions of Company C, as a successful and growing firm, the critical conflictual approach of the current research discovered and explored the covert cultural dimension of the firm. The hidden dimensions embodied mistrust and animosity of both the owners and the drivers. It characterized the conflictual organizational

culture in Company C, causing the phenomenon of rapid driver turnover. This point differed from the previous literature about taxi companies, which generally ignored the factor of internal human relationships [10,12–16,18]. One exception to this situation was the demand–control model proposed by Chungkham, Ingre, Karasek, Westerlund, and Theorell [51], which analyzed the damage of high psychological stress and the lack of ability to control decision making. These problems caused illness and undermined cultural sustainability [2,3]. This demand–control model offers beneficial insights to owners of taxi companies about how to manage their firms more productively.

The current study has contributed information about volatility in this transportation sector and its possibility of harming organizational sustainability [2,3,20,27–29,32]. This could prevent rapid turnover and provide more stability among their employees.

If the drivers protested their work conditions, the owners ignored them or responded with explicit expressions demonstrating the owners' superior power position [62]. The owners even blamed the drivers for not taking the initiative in various situations, after demanding total obedience. This analysis discovered how the owners' views and beliefs justified their system of exploitation, a justification that gave them the legitimacy to treat drivers in this manner [10,11]. This managerial behavior stemmed from the capitalist motivation [52,58] to become rich in a short time, without considering the long run destructive outcomes regarding the company's sustainability [20,27–29,32]. The bosses did not have the wisdom to understand this situation because they, themselves, had constructed this reality. The situation was a self-fulfilling prophecy; after they did not expect any initiative from their workers, the employees behaved according to this expectation [9,63,64].

This study explored why the relationship between management and employees in Company C were in conflict. The environment was stressful in Company C, the owners did not respect their employees, and the workers frequently left the firm. Although turnover was a common phenomenon in other taxi companies, the reasons were different [17,18]. Although other studies observed high levels of stress, aggressive events, and negative attitudes in other taxi companies [15], turnover was a result of a lack of a culture of safety along with high risk, which caused sickness and stress among drivers. In contrast, employees in Company C left their work because of low wages, poor work conditions, and abusive treatment. This research offered an explanation for the high worker turnover among taxi companies in general: the poor work conditions of the job and the outright exploitation of taxi drivers. Rapid worker turnover in the taxi industry can harm the industry's sustainability and cause economic losses [1,10,20,21]. This phenomenon is prevalent in Europe and in the United State because of the growth of the "gig economy", which creates more independent and autonomous taxi drivers, but whose work environment has become increasingly temporary, volatile, and economically stressed [18].

The findings in the current research strengthen the activity theory and Latour's interaction theory by emphasizing power and force relationships, which caused damage to cultural sustainability. The findings provide examples from everyday life of how social reality was structured in this organizational setting [21–25].

The prior literature discussed work conditions in the taxi industry from the perspective of the union. In many of those studies, the drivers were fighting together against new technologies, such as Uber, which had disrupted the taxi business and harmed the drivers' work conditions [13]. Other studies focused on the drivers' struggles against other types of exploitation [12]. The current study analyzed one taxi company and described the drivers' individual combat against exploitation. Drivers in Company C earned low wages. Young drivers, who were uneducated and often otherwise unemployable, did not have a lot of choices. They took driving work as the only opportunity that they had. They continued to work until they could find a better alternative. The steady, older workers stayed in the company in order to fill their time. They had free time and were willing to work even for low wages. As mentioned in the findings, this group already had pension payments from their previous work. Nevertheless, even this group suffered exploitation, and the owners knew how to manipulate each group of drivers for the company's benefit [11,12].

The current study focused on the pressure in work conditions. Previously, the literature had discussed the pressure stemming from high-risk conditions in the workplace, which caused illness and injuries [12,14,15]. The current research enriched the literature by showing how greedy owners of a taxi company could cause high levels of stress for their drivers. They demanded long hours of work, interfered with the routes, and did not pay reasonable wages. These work conditions caused the drivers' high level of anger against their bosses.

Moreover, the contribution of this ethnographic study was to supply a picture of everyday life: the rituals, patterns of behavior, norms, and the rude language that formed a routine style in management of a particular taxi company, which the prior literature had not discussed [6]. This autocratic managerial style constructed everyday life in a manner that served the needs of the owners; it gave them more control over the drivers' routine [51,52,58]. This business strategy benefited the owners in the short run, but harmed the sustainability of the firm. The drivers and other workers were abandoning the company, which caused instability and volatility for the business [3]. Using the interpretive method of Lukka [41], the current ethnographic approach analyzed the hidden norms, which became clear and explained how the owners created institutionalized routines, ritual, and habitual daily life. These hidden norms became part of the regulation of this complicated reality. This construction of regulations was very beneficial to the owners' needs and helped them to control the definition of reality [6–9,62,64]. The ownership position gave them the power to take advantage of the inferior status of the drivers. They knew that if the drivers left, they could find alternative unemployed workers quickly.

The current findings once again demonstrated taxi drivers' exploited status in the capitalistic system. Mathew [13] discussed the low wages of drivers and their exploitation by the capitalist system in New York City. In many of the same ways, the owners of Taxi Company C utilized various tactics against these marginal workers. They used many tricks to cheat the unorganized employees. Notably, the owners humiliated the drivers, using curses when they spoke with them. The owners sought to employ workers in near-slave-like conditions, in which the workers would obey every demand. The owners used a software system that helped them track and surveil their drivers. Previously, there was a paucity of literature about the systems of supervision in the taxi industry. The current research provided vivid descriptions of how the owners of the taxi firm could supervise the drivers' work, interfere with the work, and further exploit them. The capitalist system in Israel constantly provided alternative impoverished workers who could be means to their bosses' end [13]. At the time of this study, Israeli taxi drivers had neither unionized, nor had fought for their legal rights in any meaningful way [12,13]. This lack of action was in contrast to the nascent organizing efforts in Germany, Japan, and other countries; although these unions were just beginning their struggle for better working conditions [10,14,15,17].

Capitalism and globalization motivate the traditional management style; whose goal is to maximize profits [65]. This style tends to prioritize efficiency and neo-liberal market mechanisms. As a result, management exploits workers by paying low wages, maintaining substandard work conditions, and threatening termination [66,67], but alternative values, such as relationships based on cooperation and moral equality [67], could foster sustainability, avoid the excesses of capitalism, as well as maintain profitability. Although there are a number of managerial styles in these alternative organizations, the more successful ones encourage workplace democracy while contending with complicated business conditions [65–68]. The more equitable personal relationships in these alternative organizations offer a model for enhancing organizational sustainability in taxi companies.

In summary, this study has enriched the literature about the organizational culture of the taxi sector by exploring its conflictual features between an employer and his workers [1,12,35]. It has also contributed to the literature by pointing out how the role of managerial attitude has affected human interactions between employers and employees, and how these interactions could construct a culture that benefited the ruler of the organization [6–9,20,27,32,36,61]. This research stressed the connection between

a cultural interpretive perspective and organizational sustainability by presenting the covert exploitation and dishonesty in the human relationships in the working environment. Other businesses could learn from this study how to avoid an exploitive working environment. A more harmonious organizational culture would take the employees' needs into consideration, and thus enhance cultural and economic sustainability [2–4,15].

## 7. Conclusions

This ethnographic study analyzed organizational culture in one taxi firm, showing the routine and habitus of everyday life. The study exposed the exploitive and self-destructive nature of the capitalist system; how owners of the company justified their negative attitude toward their employees and executed management procedures to exploit those employees. By supplying a view of the cultural structure of that firm, this ethnographic research then analyzed the organizational pattern of behaviors, hidden norms, and the assumptions of the company. The study discovered cultural norms driven by materialistic values and beliefs. The desire to become rich quickly was the main motivation of the owners. To achieve this end, the owners utilized management tactics of unbridled capitalism, which included dehumanization and dishonesty. Ironically, the owners could not enjoy their wealth, because they too existed in the company-wide state of stress and pressure, which they had constructed. In brief, their management policies had boomeranged.

This case study analyzed work conditions at a particular taxi company. Obviously, these unfair work relationships, expressed as cultural dimensions, could also exist in other industries when managers manipulated the factor of human relationships. This avaricious drive was inherent in capitalism and the neo-liberal lifestyle, motivating organizations to become "successful" by denying employees their social rights. Thus, exploitive capitalism could act like a boomerang and harm the business. This study offered some practical implications to improve cultural sustainability in the taxi industry and also in other businesses.

The current study, analyzing the management style of one specific firm, provides a warning alarm to the existing system, which prevents honest and fair work conditions. An autocratic managerial style can harm a company by constructing a culture of mistrust and a lack of participation of organizational members. High levels of pressure and constant surveillance of employees causes greater damage than benefit for the company. In order to create a more harmonious work environment and to ensure organizational sustainability, management will need to employ moderate capitalism with fair wages. This is more successful than utilizing unbridled capitalism. It is difficult to maintain a successful business with high rates of turnover; retaining employees is a better alternative in the long run and will provide organizational sustainability. Business sustainability creates equitable work conditions and constructs stability among employees [2,3]. A culturally successful firm provides honest and fair working conditions in the long run; to ignore this can backfire and ruin the reputation of a business investor. Unions can better protect taxi drivers by studying the exploitative practices of this particular taxi company.

The distinctive contribution of this study is the awareness of harmful managerial behavior; managers need to know how to avoid negative interaction in the workplace in order to achieve sustainability. The owners in this case study did not consider managing the company with sensitivity towards human resources or using a supportive, democratic style of management [19,20,32].

This case study presented the difficulties of the taxi driving company in achieving sustainability due to the lack of balance among four elements: financial, ecological, social, and workers' well-being [32] (p. 40). The drivers suffered from the lack of a place to rest, the lack of a positive environment, the lack of financial well-being, and the lack of basic work conditions, which Galpin et al. [32] wrote about. Galpin et al. [32] stressed the importance of the four elements, which together have a cumulative interaction, promoting sustainability. Unfortunately, there were negative cumulative factors that impeded the taxi company's ability to achieve sustainability: conflicts, poor communication, oppressive control, and exploitative salaries.

The literature about sustainability stressed the role of managerial quality in regard to human resources. This case study provided an illustrative example of how the management ignored potentially constructive behavior when dealing with employees. The outcome was a stressful atmosphere, with a lot of resentment and animosity toward the owners of the company [10,29,63,69]. Moreover, the management behaved in an authoritarian way; they did not consult with their employees and they shunned a more supportive management style [19,20].

This study provides practical implementations to achieve sustainability and to avoid the troublesome behavior of the owners in this case study:

1.  Control systems and supervision over subordinates are more effective when the employees accept them. In order to achieve sustainability, management should not use extreme measures; thus, avoiding bad feelings and animosity among employees. In this case study, the management caused more damage than benefit when it used technological tools to track the drivers [21,22].

2.  An organization that wants to achieve sustainability needs to avoid the phenomenon of frequent worker turnover [12,13]. Long-term workers are more loyal and dedicated to the firm. The organization can avoid frequent worker turnover in several ways:

    a.  Providing fair salaries in comparison to other companies.
    b.  Behaving and communicating in a respectful way to subordinates, providing a place for resting, coffee breaks, and meals.
    c.  Planning reasonable work schedules, requiring no more than eight hours per day, with breaks to avoid non-stop work.
    d.  Creating a supportive and democratic management style in the organization.

The current study experienced the research limitation of focusing on only one case study. Thus, it was difficult to make generalizations about other taxi companies in Israel and other countries. Nevertheless, the findings were interesting enough to warrant future research comparing several firms in the taxi industry. This research would seek common features in the organizational culture of the taxi industry and ask how this culture correlates with economic sustainability.

**Funding:** Zefat Academic college supported this research.

**Institutional Review Board Statement:** The study was conducted in accordance with the declaration of Helsinki, and approved by Zefat academic institution 10 August 2018.

**Informed Consent Statement:** Informed consent was obtained from all subjects involved in the study.

**Data Availability Statement:** This data was collected by interviews and by direct observation, I don't any other source.

**Conflicts of Interest:** The author declares no conflict of interest.

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
