# Peer review of "Management Attitude Shaping Cultural Sustainability in a Taxi Company—An Israeli Case Study"

_sustainability, doi:10.3390/su142013109_

Round 1

Reviewer 1 Report

The article seeks to analyze the interaction of management behavior in shaping organizational culture and sustainability. From this objective, based on an exhaustive analysis of a case study, they provide an interesting perspective on labor relations that could be extrapolated to other types of companies with similar characteristics.

From a qualitative approach, the methodology, which is based on ethnography, is very well described and substantiated. Likewise, the text is interesting and is considered a valuable contribution to the study of sustainability and particularly to the cultural dimension. The text is written coherently and cohesively.

Some writing errors that are recommended to be corrected before publication are described below:

  • Check the chart as the text is out of date.

Page 1:

-Introduction misspelled.

-Line 14: Author and author (missing “and”).

Page 2: 

-Line 7: “most researches” (missing plural).

-Line 10: from a cultural perspective (missing a).

Page 3: 

-Line 2: It is not clear what “it” refers to.

Page 5:

-Line 26: “are young males (are) from” (two verbs are).

-Line 49: “It was a family business” (Wrong pronoun).

Page 6:

-Line 37: “drivers” misspelled.

Page 7:

-Line 27: “They did not want (not) to reveal” (extra not).

Page 15:

-Second paragraph: Where does the site end?

Page 16:

-4 paragraph, line 4: missing endpoint.

Page 17:

-3 paragraph: endpoint at the beginning.

Author Response

All the corections are in the attached file

Thanks yaffa

Reviewer 2 Report

This paper explores the managerial culture through a study case of a taxi company in Israel. Although the main topic is rather interesting, it needs further work before it might be accepted for publication. My main concerns are that there is no proper discussion with a deeper theoretical approach further than the taxi’s matter approaches, therefore the conclusions are not working since it is not clear what is the distinctive contribution to the sustainable science scholarly literature.

 State of the art and key literature: It is too specific and presents a literature review that is focused on taxi matters quite exclusively but this study case could go further in the existing theories about the managerial organization and the technology changes. I think that this study case could be enriched through CHAT theory or Latour’s theory. Also I suggest to explore González, V. M., Nardi, B., & Mark, G. (2009).

 Methods: This section seems well explained, however, it is missing a better explanation about the way the external perspective has been applied and how the assistant researcher has been able to produce the estrangement if he was involved previously with the firm. Also, it is suggested to increase the number of interviews with old workers from different periods throughout the company life.

 Discussion: It is missing a deeper engagement of the main concepts of the literature review with the results. Its analysis is problematic. It is needed to engage the main concepts or ideas put forward in the theory section. The reader is left with a feeling of longing in what concerns their reasoning in light of the research findings. It needs some further explanations. I felt strongly normative explanations in the text.

 Conclusion: It is needed to explain better the distinctive contribution to sustainable science scholarly literature.

 Minors mistakes: Some typing mistakes and some sentence seems to be an opinion.

Author Response

Dear reviewer,

Here is my responses to your review,

Thank you very much!

Yaffa

Reviewer 3 Report

Management attitude shaping Cultural Sustainability in a Taxi
Company an Israeli case study

Abstract

Good, but it should highlight the gap/ or the problem or the driver for the research..

The paper is very well structured and well written. The information are well organised and well presented.

The paper presents original study and contributes to the literature.

Author Response

Here is my response to your review in the attached file.

Thank you vety much

Yaffa

Round 2

Reviewer 2 Report

The abstract should be rewritten, following the introduction of a new theoretical framework. This will also have implications for associated conclusions.

 State of the Art and Key Literature: Although this section has been enriched and improved by the addition of activity theory, a clear link to the theory of the previous paper has been missed. At present, there is a lack of fluidity and connection amongst and between the theories explored.

 Methods: The comments made about this section haven’t been addressed. For instance, the explanation regarding the creation of estrangement should be improved. At the same time, the number of interviews still needs to be increased, until saturation can be demonstrated.

 Discussion: This section has been enhanced, but requires additional improvements. In a qualitative study, data is produced when analytical links are established with theories presented earlier. Chapters 4 and 5 lack this approach, creating a weakness in the paper.

Conclusion: The distinctive contribution to sustainable science scholarly literature is not obvious, and still needs to be demonstrated.

Author Response

Dear reviewer

His my responses

Round 3

Reviewer 2 Report

The author has integrated all the commentaries. The paper has improved notably. I think it is ready for publication.